# A Decellularized Human Limbal Scaffold for Limbal Stem Cell Niche Reconstruction

**DOI:** 10.3390/ijms221810067

**Published:** 2021-09-17

**Authors:** Naresh Polisetti, Benjamin Roschinski, Ursula Schlötzer-Schrehardt, Philip Maier, Günther Schlunck, Thomas Reinhard

**Affiliations:** 1Eye Center, Medical Center, Faculty of Medicine, University of Freiburg, Killianstrasse 5, 79106 Freiburg, Germany; benjamin.roschinski@uniklinik-freiburg.de (B.R.); philip.maier@uniklinik-freiburg.de (P.M.); guenther.schlunck@uniklinik-freiburg.de (G.S.); thomas.reinhard@uniklinik-freiburg.de (T.R.); 2Department of Ophthalmology, University Hospital Erlangen, Friedrich-Alexander-University of Erlangen-Nürnberg, Schwabachanlage 6, D-91054 Erlangen, Germany; Ursula.Schloetzer-Schrehardt@uk-erlangen.de

**Keywords:** decellularization, limbal tissue engineering, limbal transplantation, ex vivo transplantation, recellularization, decellularized limbal tissue, limbal stem cells, limbal melanocytes

## Abstract

The transplantation of ex vivo expanded limbal epithelial progenitor cells (LEPCs) on amniotic membrane or fibrin gel is an established therapeutic strategy to regenerate the damaged corneal surface in patients with limbal stem cell deficiency (LSCD), but the long-term success rate is restricted. A scaffold with niche-specific structure and extracellular matrix (ECM) composition might have the advantage to improve long-term clinical outcomes, in particular for patients with severe damage or complete loss of the limbal niche tissue structure. Therefore, we evaluated the decellularized human limbus (DHL) as a biomimetic scaffold for the transplantation of LEPCs. Corneoscleral tissue was decellularized by sodium deoxycholate and deoxyribonuclease I in the presence or absence of dextran. We evaluated the efficiency of decellularization and its effects on the ultrastructure and ECM composition of the human corneal limbus. The recellularization of these scaffolds was studied by plating cultured LEPCs and limbal melanocytes (LMs) or by allowing cells to migrate from the host tissue following a lamellar transplantation ex vivo. Our decellularization protocol rapidly and effectively removed cellular and nuclear material while preserving the native ECM composition. In vitro recellularization by LEPCs and LMs demonstrated the good biocompatibility of the DHL and intrastromal invasion of LEPCs. Ex vivo transplantation of DHL revealed complete epithelialization as well as melanocytic and stromal repopulation from the host tissue. Thus, the generated DHL scaffold could be a promising biological material as a carrier for the transplantation of LEPCs to treat LSCD.

## 1. Introduction

Homeostasis of the corneal epithelium, which is an important prerequisite for corneal transparency and visual function, is maintained by the reservoir of limbal epithelial stem/progenitor cells (LEPCs) located in the basal epithelial layer of the corneoscleral limbus [1]. The corneoscleral limbus acts as an immunologic and physiological barrier between the sclera and cornea. It has a unique architecture that comprises limbal crypts between palisades of Vogt, focal stromal projections at the corneal edge of the limbus, and a distinct vascular network [2,3]. The LEPC niche is a specialized protective anatomic location that regulates LEPC function through their interactions with extracellular matrix (ECM) components and neighboring cells via direct contact or soluble growth and signaling factors [4,5,6]. Inflammatory diseases or trauma such as an alkaline burn are the most frequent causes for LEPC loss and destruction of the distinct limbal niche tissue structure, leading to limbal stem cell deficiency (LSCD) and loss of vision [7,8]. The transplantation of ex vivo expanded LEPCs or oral mucosal epithelial cells on amniotic membrane or fibrin gel is an established therapeutic strategy to regenerate the damaged corneal surface [9,10]. It is applied in various clinical centers with a reported overall success rate of 60 to 70% [11,12,13], but long-term results beyond 10 years of follow-up are not well documented. The long-term clinical success depends on the maintenance and long-term survival of LEPCs in the graft, which reconstitutes a functional stem cell pool [5]. However, standard culture techniques largely disregard the subtleties of the niche microenvironment, which is mainly formed by the ECM and regulates stem cell homeostasis in vivo [14]. Moreover, in cases of allogenic limbal transplantation, allografts require systemic immunosuppressive treatment as the risk of transplant rejection limits the long-term prognosis for graft survival [15]. Therefore, current research focuses on identifying limbal niche components and developing appropriate biomimetic scaffolds with low immunogenic potential to replicate the biological niche in vitro and to avoid immune responses [16,17].

Decellularized tissues or organs, which possess tissue-specific three-dimensionality, have long been advocated as cell-free scaffolds of intact ECM for subsequent cellular repopulation in tissue engineering and clinical applications [18,19]. The decellularized human limbus (DHL) may provide a supportive natural scaffold for LEPC expansion and transplantation. Donor corneas not suitable for transplantation due to low endothelial cell counts or donor tissue remaining after lamellar Descemet’s membrane endothelial cell keratoplasty (DMEK) can be used for decellularization [20,21]. Various protocols have been devised to decellularize the central human cornea [22,23], but only limited data are available on the decellularization of the delicate human limbal stem cell niche [20,21]. Most protocols suggested for decellularization are very time consuming (~4–7 days) [20,21]. In a previous study, we reported on a fast and efficient method (~1 day) to decellularize the human cornea using sodium deoxycholate (SD), allowing for the preservation of mechanical, optical, and biochemical properties as well as the cellular biocompatibility of the central cornea (~6 to 7 mm diameter central cornea) [24]. These corneal scaffolds may serve to reconstruct lamellar structural defects of the central cornea. In contrast, the current manuscript aims to provide a scaffold for limbal stem cell transplantation with the long-term goal to treat patients suffering from structural damage in the limbal stem cell niche. The limbal stem cell niche is very different from the central cornea [2,3,25] studied in the previous report [24] as it is endowed with crypt structures, fine stromal protrusions, a particular ECM composition, and a basement membrane (BM) composition, which regulates LEPC homeostasis [26,27]. In light of these major structural and biological distinctions, a focused assessment of a decellularization protocol in limbal tissue is warranted. Following the repopulation of a DHL scaffold, the resulting phenotype of human LEPCs has only been examined in a single study [21]. No report has addressed repopulation with limbal melanocytes (LMs), which are found in close proximity to LEPCs at the healthy limbus [4,28] and which protect LEPCs from UV damage by transferring melanin granules. Moreover, LMs control LEPC homeostasis in the limbal niche both in vitro and in vivo and have effective immunomodulatory and anti-angiogenic properties [28,29,30,31]. In a rat limbal injury model, a decellularized limbal graft was repopulated by host cells after transplantation, supported epithelialization, and reduced corneal haze formation [21]. However, the repopulation of grafted decellularized limbal scaffolds by human limbal host cells has not been documented so far.

In the current study, we extended our previous protocol to decellularize the human corneal limbus and evaluated decellularization efficiency. We explored potential effects of decellularization on ECM components and tested the biocompatibility of the generated DHL scaffolds by seeding cultured LEPCs and LMs. We also evaluated cell migration from the host tissue by ex vivo transplantation of DHL scaffolds.

## 2. Results

### 2.1. Efficiency of Decellularization

The efficiency of limbal tissue decellularization was initially screened by hematoxylin and eosin (H&E) staining to assess cellularity. Sections of normal human limbus (NHL) showed a multilayered epithelium with darkly stained cell clusters in the basal layer of the limbal epithelium (Figure 1A, yellow dotted circles) and stromal cells (Figure 1A). Sections of DHL in the presence or absence of dextran were cell-free and the collagen fibrils of the ECM were still arranged in a regular pattern (Figure 1A). The decellularized limbal tissue was further screened for human leukocyte antigen (HLA-ABC) to detect any residual cellular/membranous material. For the detection of nuclear debris, deoxyribonucleic acid (DNA) content was measured (Figure 1B) and sections were stained with 4′,6-diamidino-2-phenylindole (DAPI, Figure 1A). HLA-ABC and DNA were not detected in DHL specimens, which confirmed the absence of any cellular or nuclear material (Figure 1A). No differences were noticed with respect to residual cellular or nuclear material when decellularization was performed in the presence or absence of dextran. Spectroscopic analysis of DNA content showed that SD and DNAse treatment removed about 98.5 ± 0.3% of the DNA content in the presence of dextran and 99.2 ± 0.4% in its absence (Figure 1B). Western blot analysis did not show any HLA-ABC expression (~41 kDa) in DHL samples, whereas its expression was clearly noted in the NHL scaffolds (Figure 1C). An uncropped version of the Western blot is shown in Appendix A.

### 2.2. Limbal Architecture by Light and Electron Microscopy

H&E staining of DHL specimens showed the presence of Bowman’s layer in the peripheral corneal region of the DHL scaffold (Figure 2A, arrow). The stroma of DHL was intact and showed a regular lamellar arrangement of the collagen fibrils (Figure 2A). Connective tissue protrusions (black arrowheads), invaginations (white arrowheads), and vascular gaps in the ECM (dashed circles) were apparent in DHL (Figure 2A). The effect of decellularization treatment on the ultrastructure of limbal tissue was evaluated by transmission electron microscopy (TEM). TEM analysis showed the stromal projections in both DHL samples with or without dextran (Figure 2B). Epithelial basement membrane (BM; arrowheads) could be clearly seen in DHL samples and more prominently visualized in dextran-treated DHL samples (Figure 2B). Collagen fibrils of DHL stroma were regularly arranged similar to NHL and did not show any significant structural abnormalities (Figure 2B).

### 2.3. ECM Components

Periodic acid Schiff (PAS) and alcian blue (AB) staining of limbal scaffolds showed the presence of glycoproteins and glycosaminoglycans (GAGs) throughout the limbal stroma. Both PAS and AB staining appeared weaker in DHL scaffolds compared to NHL (Figure 2C), whereas the amount of sulfated GAG (sGAG) content measured in a 1,9-dimethyl methylene blue (DMMB) assay was unaffected by decellularization (Figure 2D).

The ECM composition of the limbal scaffolds was evaluated by immunostaining for various components including agrin, collagens (Col III, IV, XVIII), fibronectin (FN), junctional adhesion molecule C (JAM-C), tenascin C (TN-C), vitronectin (VN), and laminin (LN) chains (α3, α5, β2, β3, and γ2). Agrin, a heparan sulfate proteoglycan, was expressed in the BM with moderate intensity in all samples (Figure 3, arrow). The expression of Col III, a fibrillar collagen, was detected in the limbal stroma, whereas non-fibrillar collagens Col IV and Col XVIII were clearly noted in the BM of both NHL and DHL samples (Figure 3, arrow). The glycoproteins FN and vitronectin were clearly detected in BM in all samples without any significant differences (Figure 3, arrow). TN-C expression was observed in the BM zone of the limbal surface in all samples, whereas JAM-C expression was localized in the limbal epithelial BM and sub-epithelial matrix as well (Figure 3, arrow). The LN chains (LN-α3,-α5, -β3 -β3, -γ2) were found to be strongly expressed in the limbal BM of DHL in the presence or absence of dextran, similar to NHL (Figure 3, arrow).

### 2.4. In Vitro Recellularization

The biocompatibility of the DHL to support LEPC and LM was evaluated. After the seeding of LEPCs and one week of cultivation, H&E staining showed a monolayer of epithelial cells on DHL-LEPC scaffolds (Figure 4A, 1 w). Stratification (2–3 layers) of the epithelium was confirmed by H&E staining in both DHL–LEPC and DHL–LEPC/LM scaffolds after 3 weeks of cultivation (Figure 4A, 3 w). Besides epithelial and melanocyte repopulation of the scaffolds, the most striking observation was the presence of epithelial-like cells in the limbal stroma (Figure 4A, 3 w, arrows).

Phenotypic characterization of recellularized scaffolds by immunohistochemical staining confirmed pronounced pan-cytokeratin (pan-CK) and intercellular E (epithelial)-cadherin (red) staining in all epithelial layers (Figure 4B, dashed line represents BM); CK15 and p63 staining were observed in basal layers (Figure 4B, red, dotted line separates basal and suprabasal cells); more Ki-67^+^ cells were observed in the basal limbal region compared to the corneal region (Figure 4B, arrows); and vimentin staining was observed in basal layers and also in the limbal stroma (Figure 4B, dotted line separates basal and suprabasal cells). Melan-A^+^ (red) melanocytes were interspersed in the epithelial layers of the DHL–LEPC/LM scaffolds (Figure 4B, arrowhead). Double immunofluorescent staining of DHL–LEPC/LM scaffolds showed the multilayered epithelium (PAN-CK^+^, red) and intermingled melanocytes (Melan-A^+^, green). The corneal epithelial differentiation marker CK3 (green) was expressed in superficial epithelial layers, whereas the melanocytes (Melan A^+^, red) associated with the basal cells (Figure 4C). TEM studies showed the 5–6-layered epithelium in both DHL–LEPC (Figure 4D(i)) and DHL–LEPC/LM scaffolds (Figure 4D(ii), dotted line separates epithelium and stroma). The DHL–LEPC/LM scaffolds showed the presence of melanocytes between the epithelial cells (Figure 4D(iii), arrow) and the presence of melanosomes in the cytoplasm of epithelial cells (Figure 4D(iv,v), arrows).

To verify the characteristics of the aforementioned epithelial-like cells within the limbal stroma, immunohistochemical analysis was performed using epithelial (pan-CK, E-cadherin, p63) and mesenchymal markers (vimentin). Pan-CK^+^, E-cadherin^+^, and p63^+^ epithelial cells were seen to invade the limbal stroma, whereas no invasion of LEPCs occurred in the corneal region (Figure 4E). Furthermore, some LEPCs invading the limbal stroma lost cytokeratin (pan-CK) and E-cadherin expression (Figure 4E, arrowheads) and underwent an epithelial–mesenchymal transition (Figure 4E, Vimentin^+^ cells, slender morphology, arrowheads). TEM confirmed the presence of stromal cells of variable phenotypes showing either epithelial features, such as roundish shape, round nuclei, and vacuolar inclusion bodies (Figure 4F(i)), mixed epithelial–mesenchymal features, such as spindle shape, keratin filaments, desmosomes, and rough endoplasmic reticulum (Figure 4F(ii)), or fibroblastic features, such as prominent rough endoplasmic reticulum (Figure 4F(iii)), indicating epithelial–mesenchymal transition. To illustrate further intrastromal invasion of LEPCs, we investigated BM components of recellularized scaffolds. Surprisingly, fibronectin, Col IV, and LN α3 expression was largely intact with occasionally distinct gaps (Figure 4G, arrows). The expression of these molecules in the BM of the corneal region was intact (data not shown).

### 2.5. Ex Vivo Transplantation

The repopulation of DHLs was also tested in a lamellar ex vivo transplantation approach to allow the migration of cells from normal host tissue. For that purpose, an anterior lamella of DHL tissue (4 mm graft) was sutured onto a posterior lamellar bed of NHL tissue (Appendix A). After 3 weeks in organ culture, H&E staining revealed the complete epithelialization of the grafted tissue with epithelial stratification (Figure 5A(i); dotted line separates the graft and host tissue). The graft comprised parts of peripheral cornea (arrow pointing the end of Bowman’s layer of the graft) (Figure 5A(ii)) and limbus (Figure 5A(ii,iii)). The graft tissue was also repopulated by stromal cells of the host (Figure 5A(ii,iii), black arrowheads). Immunohistochemical analyses revealed the expression of the epithelial marker cytokeratin (pan-CK) in the epithelial cells on the host (corneal region) and graft tissue (Figure 5B, red). Pan-CK^+^ cells were also observed in the limbal stroma (Figure 5B, arrow). The corneal differentiation marker CK3 was present only in the superficial epithelial cells of the graft and in all epithelial layers of the host tissue, with slightly pronounced staining in the superficial epithelial layers (Figure 5B, dotted line separates basal and suprabasal epithelium). The expression of the epithelial progenitor marker CK15 was apparent in basal epithelial cells on the grafted scaffold and no expression was observed in the host corneal tissue (faint staining can be seen) (Figure 5B, arrowheads). Another epithelial progenitor marker, p63, was also observed in the basal cells of the graft and in very few cells of the host tissue (Figure 5B, red, arrowheads). Moreover, we also noticed Ki-67^+^ cells in the basal layer of the epithelium covering the DHL tissue (Figure 5B, arrowheads), but no Ki-67^+^ cells were detected in the host tissue. Interestingly, we observed Melan-A^+^ melanocytes (red) in association with basal epithelial cells in the corneal region of host tissue and signs of melanocyte migration to the graft region (Figure 5B, arrowheads). Vimentin^+^ cells were present in the basal epithelial layer (arrows) and limbal stroma of the graft (arrowheads) similar to corneal stroma of the host (Figure 5B, arrowheads).

## 3. Discussion

The transplantation of ex vivo expanded LEPCs or non-ocular cells on amniotic membrane or fibrin gel is an established therapeutic strategy to treat LSCD [9,10]. However, the long-term success rate is limited and the estimated graft survival rate, i.e., survival of the transplanted epithelium, decreases from 100% at 3 years to 71% at 5 years [32,33]. In cases of structural tissue damage, a lack of supporting limbal stem cell niche components may limit the long-term viability of transplanted LEPCs [32,34]. Recently, research has focused on generating scaffolds similar to the tissue-specific niche environment to enhance stem cell survival and regenerative potential [35,36]. In cases of allogenic limbal transplantation, immunoresponse-related damage to the transplanted limbal tissue is a crucial factor for chronic graft failure, and most patients require immunosuppression [15]. Therefore, there is an unmet need for developing a scaffold that more closely mimics the limbal niche and exhibits very low immunogenicity to better maintain LEPCs during in vitro cell expansion and minimize immune response in the host. Decellularized scaffolds, which possess a tissue-specific three-dimensional architecture, gained interest as alternative carrier materials for stem cell transplantation. Therefore, it was our goal to generate DHL scaffolds from cadaveric peripheral corneal tissue with maximum decellularization efficiency, preserved limbal tissue architecture, and intact functional properties of ECM, to serve as a substrate for LEPC transplantation to treat LSCD in the future.

Various protocols have been used to decellularize human corneal tissue [22,23,24], but are limited to very few studies on human limbal tissue so far [20,21]. In these studies, combinations of SD or hypertonic salt (NaCl) and nucleases have been used to decellularize the human corneal limbus. However, significant drawbacks were the reported long elution times required for the removal of cellular components (~4–7 days). In the current study, we validated our recently published rapid decellularization protocol [24] for efficient decellularization of limbal tissue. Our data show that 1% SD with DNAse and 4% dextran allows for fast (~1 day) and efficient decellularization of limbal tissue with complete removal of cellular components (HLA-ABC and nuclear debris). Thus, the rapid decellularization protocol can be extended to treat limbal tissue with similar efficiency as corneal scaffolds [24]. The lack of nuclear and cellular material in decellularized tissues reduces the antigenicity of the limbal scaffolds and could therefore provide nonimmunogenic scaffolds for LEPC transplantation to treat LSCD.

The structural architecture of the limbus provides a repository for LEPCs, which are essential for the long-term homeostasis of the corneal epithelium. The structure of the limbus comprises limbal crypts between the palisades of Vogt and focal stromal projections at the corneal edge of the limbus, extending from the palisades in a finger-like pattern [2,3]. The current study shows that DHL contains typical limbal connective tissue projections and invaginations, suggesting the preservation of the limbal architecture during decellularization. The human limbus is also characterized by a distinct vascular network with radially arranged arteries and veins [2]. Similarly, the DHL preserved radially oriented vascular channels, which may support revascularization and nutrition after transplantation. It has been reported that sodium dodecyl sulfate (SDS) disrupted the ultrastructure of the cornea most severely, resulting in the disorganization of collagen fibrils in the stroma of the cornea and limbus [37]. Similar results have been reported for SDS-treated corneas, which had the greatest loss of transparency with most ECM disruption [23]. Significant ECM disruption or collagen damage was not observed in the present study, suggesting the preservation of ECM architecture similar to an earlier study on DHL [20].

The fate of stem/progenitor cells is mainly regulated by the niche microenvironment with its distinct ECM components and mechanical properties, which play a key role in cell adhesion, proliferation, and migration [38]. Therefore, it is crucial to preserve the integrity of ECM and BM components in the decellularized scaffolds for further tissue engineering and clinical applications. The limbal tissue has a specialized BM and ECM protein composition, which contributes to the maintenance of LEPC phenotype and influences the amount and composition of cytokines and growth factors acting on LEPCs [25,39,40]. Previous studies on decellularized limbal tissue provided little detailed examinations of ECM components [20,21]. Agrin, a heparin sulfate proteoglycan that promotes LEPC proliferation and corneal wound healing [41], is highly expressed in the limbal region and its expression declines towards the central cornea [25]. Agrin was preserved in the DHL scaffolds of the current study. Col IV is one of the most abundant BM molecules in most stem cell niches including the limbal stem cell niche [40,42], and it was clearly present in the limbal region of DHL tissue. JAM-C, another limbus-specific ECM [40] that has been suggested to play a role in the migration and adhesion of hematopoietic cells and vascular endothelial cells to the limbal niche [43], was localized to the limbal BM and sub-epithelial matrix of DHL. We reported previously that the LN chains α2, α3, α5, β1, β2, β3, γ1, γ2, and γ3 are strongly expressed in the limbal BM and that the α5-containing isoforms LN-521 and LN-511 enabled efficient expansion of both LEPCs and LMs [28,44]. The current study showed the preservation of LN chains (α3, α5, β2, β3, and γ2) in decellularized limbal scaffolds. It has been reported that ex vivo expanded allogenic LEPCs were not observed on the ocular surface 3 months after transplantation [45]. In contrast, allogenic cells were evident for up to 3 years after a keratolimbal allograft [46]. These findings illustrate the importance of LEPC niche components for long-term LEPC survival. The preservation of limbus-specific ECM (Col III, TNC, JAMC) and BM components (LN-α5 and -β2) observed in the current study is suited to support the adhesion, proliferation, and long-time survival of LEPCs.

The functional properties of DHL scaffolds can be determined by their ability to act as a niche for stem cell homeostasis, which includes cell adhesion, proliferation, migration, and differentiation of LEPCs. We explored different strategies for DHL repopulation by plating cultured limbal cells or by allowing cells to migrate and spread from adjacent host tissue. In both approaches, the limbal scaffolds were successfully repopulated and maintained a specific limbal phenotype, as characterized by the expression of progenitor markers (CK15, p63, vimentin rather used as mesenchymal marker in this study) in basal layers; the differentiation marker CK3 in superficial layers [47]; and more Ki-67^+^ cells in the limbal scaffolds compared to corneal scaffolds, similar to a previous study [22]. It has been stated that a certain number of p63^+^ (more than 3%) in the donor tissue is required for the successful regeneration of the host corneal surface [10]. Our data showed the preservation of p63^+^ LEPCs throughout the limbal basal layer (Figure 4B and Figure 5B), suggesting the potential application of these scaffolds for corneal surface regeneration. Stromal repopulation is a slower process and a time frame of 6 months was reported for complete corneal stromal repopulation in keratoconus patients after collagen crosslinking treatment [48]. In rats, it has been reported that no stromal infiltration in decellularized limbal grafts was apparent within one week of transplantation [22]. In our previous publication, we reported some stromal repopulation of a decellularized cornea after 5 weeks of lamellar corneal transplantation (~8 mm corneal tissue) in an ex vivo model [24]. Similarly, the present study revealed a repopulation of decellularized limbal stroma by host stromal cells within 3 weeks. Previous studies highlighted the ability of decellularized limbal scaffolds to support LEPCs [22]. In this study, LMs were also included, which are found in close proximity to LEPCs and play an important role in LEPC homeostasis both in vitro and in vivo [28,29,30,31]. Our in vitro cell plating studies indicate that LMs intermingle with LEPCs on DHL scaffolds and transfer melanosomes to surrounding LEPCs (TEM studies), suggesting the potential of the scaffold to facilitate LM survival and function. Ex vivo lamellar graft experiments clearly showed the migration of LMs from donor limbal tissue onto decellularized limbal grafts, which is in line with a role of LMs in ocular surface regeneration in vivo [30,49]. The appearance of LMs on transplanted limbal scaffolds supports the recent clinical finding of the reappearance of limbal pigmentation after 8 months of a simple limbal epithelial transplant [50]. So, LMs act as emerging key players in the niche regulation of LEPCs, especially in wound healing [31]. All these observations indicate that the DHL scaffold provides a niche microenvironment to maintain the limbal phenotype of LEPCs and functional properties of LMs. Thus, DHL scaffolds provide a promising experimental model to investigate limbal niche cell interactions and may serve as an ideal scaffold for future clinical applications. However, in vivo studies are necessary to determine immunogenicity and in vivo biocompatibility.

Another striking observation was the intrastromal invasion of LEPCs in decellularized limbal scaffolds. Intrastromal invasion of LEPCs has been reported in limbal explant culture models [51,52] and on intact amniotic membrane [53]. In the present study, the invaded LEPCs lost expression of epithelial markers (pan-CK and E-cadherin), acquired spindle-shaped morphology, and expressed vimentin. Similar observations were reported in limbal explant culture models [51,53]. Surprisingly, the BM proteins type IV collagen and LN-α3 were focally absent, suggesting that intrastromal invasion happened by the dissolving of BM. This observation is similar to an earlier report on the dissolution of BM proteins of intact amniotic membranes after the culturing of limbal cell explants for 2 weeks [53]. Further studies are warranted to better characterize the mechanism of invasion and EMT, which might imply how EMT can be involved in the fate decision of LEPCs between regeneration and fibrosis during wound healing.

In conclusion, we generated decellularized limbal scaffolds in a fast and efficient manner using clinically applicable SD with DNAse in the presence of dextran. The limbal tissue architecture and ECM composition of the limbal niche were well preserved in DHLs. The DHL scaffolds revealed excellent biocompatibility for cultured LEPCs and LMs, and ex vivo transplanted limbal tissues were completely epithelialized with interspersed melanocytes as well as stromal repopulation from host tissue. Hence, a decellularized limbal graft provides a limbal niche-specific microenvironment and could be a promising scaffold to transplant LEPCs for the treatment of LSCD.

## 4. Materials and Methods

### 4.1. Tissue

Organ-cultured human corneoscleral tissues (*n* = 136) were obtained from the LIONS Cornea Bank Baden-Württemberg, Eye Center, University Medical Center Freiburg, (Freiburg, Germany). Informed consent for research use of remnant tissue had been given by the donors or their next of kin. For decellularization, remnants after DMEK (*n* = 70) and corneal buttons unsuitable for transplantation (*n* = 37) were used (Table 1). To isolate primary limbal cells for repopulation experiments, organ-cultured corneoscleral rims remaining after penetrating keratoplasty were used (*n* = 29, Table 1). The study was approved by the institutional review board of the University of Freiburg (25/20) and followed the tenets of the Declaration of Helsinki. No organs or tissues from prisoners were used.

### 4.2. Decellularization

For decellularization, we used whole corneoscleral tissue to simplify handling and avoid structural damage [37]. Decellularization was carried out as published previously [24]. Briefly, corneoscleral tissue was washed in Dulbecco’s phosphate-buffered saline (DPBS; 3 × 5 min), placed in 12-well plates with 1%, SD in ultrapure water for 30 min, and then rinsed in DPBS (3 × 30 min). Subsequently, the tissues were incubated in DNAse I, 1 mg/mL in DPBS (Roche, Mannheim, Germany) overnight under a sterile hood and terminally washed in DPBS (4 × 30 min). Corneoscleral tissues without any further processing were used as controls (Normal Human Limbus; NHL). For decellularization, all washing steps and SD (Sigma-Aldrich, Hamburg, Germany) incubation were carried out under continuous agitation (800 rpm) at room temperature. To see the effect of dextran on limbal tissue during decellularization, all decellularization steps were carried out in presence of 4% dextran. The whole decellularization procedure was carried out under aseptic conditions to ensure tissue sterility. After decellularization, the corneoscleral tissue was cut centrally with an 8 mm trephine and peripherally with a 13 mm trephine, generating a limbo-scleral tissue ring of 5 mm width, which comprises regions of the peripheral cornea (~2 mm wide) and sclera (~2 mm wide) to ensure the limbus (~1 mm wide) was included (Appendix A). Non-decellularized control tissue was cut in a similar fashion. These preparations of normal corneoscleral tissue are referred to as normal human limbus (NHL) and of decellularized corneoscleral tissue as decellularized human limbus (DHL), respectively. NHL and DHL were used for all subsequent experiments unless stated otherwise.

### 4.3. Histology

For routine histology, tissue was either fixed in 4% paraformaldehyde (30 min) and embedded in paraffin or embedded and frozen in optimal cutting temperature (OCT) medium. Five µm thick sections were cut and stained as described previously [24]. Briefly, sections were stained with hematoxylin (Haematoxylin Gill III, Surgipath, Leica, Germany) for 2 min and 1% eosin Y (Surgipath, Leica, Germany) for 1 min to observe the gross tissue architecture and degree of decellularization. To visualize the proteoglycan content, cryosections were stained with 1% alcian blue (Morphisto, Offenbach am Main, Germany) for 30 min together with a counterstain of nuclear fast red (Morhphisto, Offenbach am Main, Germany). For glycoproteins, PAS staining was performed using 1% periodic acid (Honeywell Fluka, Charlotte, NC, USA) for 10 min, and Schiff reagent (Roth, Karlsruhe, Germany) for 90 s. Samples were examined using either a Hamamatsu NanoZoomer S60 (Hamamatsu Photonics, Herrsching, Germany) or a bright field fluorescence microscope (Axio Imager.A1, Zeiss) and images were processed using ProgRes CapturePro Software (JENOPTIK, Dreseden, Germany)).

### 4.4. Immunostaining of Frozen Sections

Immunostaining of frozen sections was performed as previously described [24]. Briefly, human corneal scleral tissue in optimal cutting temperature medium was cut into 10 µm sections, fixed in 4% paraformaldehyde (PFA) for 20 min or acetone for 10 min, followed by permeabilization in 0.3% Triton X-100 in PBS for 10 min. The sections were blocked with 10% normal goat serum (NGS) and incubated with primary antibodies (Appendix A) diluted in 1% NGS in PBS overnight at 4 °C. Fluorescein isothiocyanate-conjugated or rhodamine-conjugated anti-mouse or -rabbit immunoglobulins (Life Technologies, Carlsbad, CA, USA) were used for detection, and nuclear staining was performed with DAPI (Vectashield antifade mounting medium with DAPI; Vector, Burlingame, CA, USA). Immunolabeled cryosections were examined with a fluorescence microscope (Olympus BX51; Olympus, Hamburg, Germany) or a laser scanning confocal microscope (TCS SP-8, Leica, Wetzlar, Germany). For negative controls, the primary antibodies were replaced by equimolar concentrations of an irrelevant isotypic primary antibody of the same species.

### 4.5. Immunostaining of Paraffin Sections

Immunohistochemistry was performed as previously described [54]. The list of antibodies is provided in Appendix A.

### 4.6. Transmission Electron Microscopy

For transmission electron microscopy (TEM), tissue specimens were processed as described previously [55]. Briefly, the samples were fixed in 2.5% glutaraldehyde in 0.1 M phosphate buffer, dehydrated, and embedded in epoxy resin according to standard protocols. Ultrathin sections were stained with uranyl acetate–lead citrate and examined with an electron microscope (EM 906E; Carl Zeiss Microscopy, Oberkochen, Germany).

### 4.7. DNA Content

For extraction of DNA, DNeasy Blood & Tissue Kit (69504, Qiagen, Hilden, Germany) was used. Both NHL and DHL scaffolds were incubated with proteinase K in ATL buffer at 56 °C under continuous agitation and further processed for DNA elution as per the manufacturer’s protocol. The samples were analyzed photometrically at 260 nm wavelength using the NanoDrop OneC Microvolume UV–Vis spectrophotometer (Thermo Scientific, Dreieich, Germany).

### 4.8. Sulfated Glycosaminoglycans

The sGAG content of DHL was determined using a DMMB assay (Proteoglycan detection kit, amsbio) according to the manufacturer’s protocol. Both NHL and DHL scaffolds were digested with papain as per manufacturer’s protocol (Tissue digestion kit, amsbio) and analyzed as described previously [24]. Briefly, the tissue was homogenized and digested with papain at 60 °C for one hour, when acetic acid and Tris-hydrochloric acid were added. The digested samples were added to DMMB and absorbance was measured at 515 nm using a spark microplate reader (TECAN). The absorbance was expressed in percentage (%) with reference to NHL (100%).

### 4.9. Cell Culture

LEPCs and LMs were isolated, cultivated, and characterized as described earlier [24].

### 4.10. Repopulation of Decellularized Limbal Scaffolds with Cultured Cells

For DHL–LEPC scaffolds, human LEPCs (P1) were seeded (5 × 10^5^ cells/scaffold) on the anterior surface of the DHL scaffold (either full tissue or quarters of DHL scaffold, Appendix A) and cultured initially in serum-free keratinocyte medium supplemented with bovine pituitary extract, epidermal growth factor (Life Technologies, Carlsbad, CA, USA), and 1× penicillin–streptomycin mix (Pan Biotech, Aidenbach, Germany). After three days, the medium was switched to the corneal culture medium (CCM) containing DMEM/Ham’s F12 (3:1) (Hyclone; GE Healthcare Life Sciences, Freiburg, Germany) supplemented with human corneal growth supplement (Gibco, Thermofisher Scientific, Karlsruhe, Germany), 5% FCS (GE Healthcare Life Sciences, Freiburg, Germany), and a low calcium concentration (0.4 mM Ca^2+^, labeled CCM-low). For DHL–LEPC/LM scaffolds, both LEPCs (5.0 × 10^5^ cells/scaffold) and LMs (1.5 × 10^5^ cells/scaffold) were seeded together in a ratio of 3:1 on the decellularized limbal surface and cultured as mentioned above. After one week of culture, the tissues were raised to the air–liquid interface for stratification of DHL–LEPC and LEPC/LM scaffolds and the CCM was switched to high calcium concentrations (2.0 mM Ca^2+^, labeled CCM-high) and cultured further for 2 weeks. All cultures were maintained at 37 °C, 5% CO_2_, and 95% humidity and the medium was changed every other day. After terminating the cultivation, limbal scaffolds were fixed for immunohistochemistry or light and electron microscopy as described above.

### 4.11. Ex Vivo Transplantation

The feasibility of the DHL to be used as a scaffold for limbal transplantation was tested by performing anterior lamellar ex vivo transplantation on NHL (host) (Appendix A). For lamellar preparation, host donor corneoscleral tissue was trephined to ~50% depth with a 4 mm trephine at the peripheral corneal limbal region, and the anterior host tissue was removed using a surgical knife to make a lamellar limbal excision at each site to create an inlay bed as a partial limbal defect model for the grafts. For graft preparation, DHL was processed similar to the host tissue. Each graft contained approximately 1.5 mm of peripheral cornea and a 1.5 mm portion of the sclera to ensure that the limbus (~1 mm) was included. The prepared grafts were sutured into the previously prepared 4 mm host recipient beds with three single stitches in triangle shape using polyamide suture material (10-0 Ethilon, ETHICON, Johnson & Johnson Medical Devices, Cincinnati, OH, USA). Thereafter, the corneal samples were transferred into 12-well plates and cultured in a CCM-high medium for 3 weeks.

### 4.12. Statistics

Statistical analyses were carried out as described previously [24]. Briefly, statistical analyses were performed using GraphPad Prism software (Version 6.0; Graphpad Software Inc., La Jolla, CA, USA). Data are represented as mean ± standard deviation (S.D.) from individual experiments (Table 1) or as mean ± standard error of the mean (S.E.M.) (graphs). The statistical significance (*p* ≤ 0.05) was evaluated by the Wilcoxon signed-rank test or Mann–Whitney U test.

## Figures and Tables

**Figure 1 ijms-22-10067-f001:**
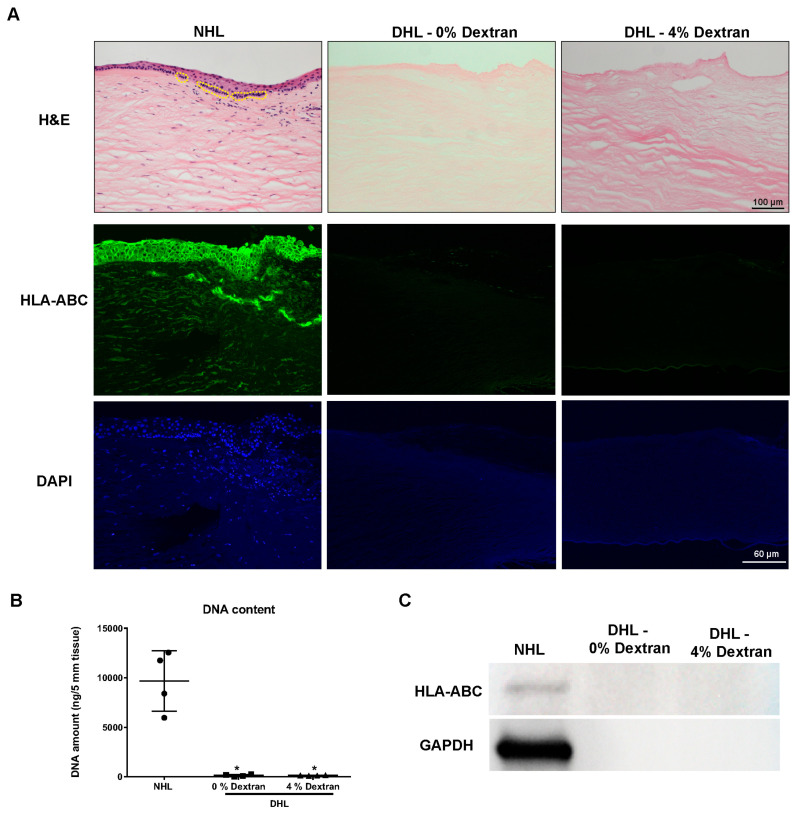
Decellularization efficiency. (**A**) Histological evaluation of human corneal limbus decellularized by sodium deoxycholate (SD) with deoxyribonuclease I in 0 or 4% dextran, in comparison to the normal human limbus (NHL) using hematoxylin and eosin (H&E) staining. The NHL sections show a multilayered epithelium with darkly stained cell clusters (yellow dotted circle) and stromal cells, whereas decellularized human limbus (DHL) sections show no cell remnants. HLA-ABC staining (green) of NHL and DHL showing cellular expression of HLA in NHC; No HLA expression in DHL scaffolds. DAPI staining reveals nuclei in NHL without remaining nuclei/nuclear debris in DHL. (**B**) Confirmation of decellularization by quantification of residual DNA. The graph represents the quantity of DNA in both NHL and DHL (0 or 4% dextran). Data are expressed as means ± S.E.M. (*n* = 4). * *p* < 0.05; Mann–Whitney U test. (**C**) Western blot showing the band of GAPDH and HLA-ABC in the NHL; No bands in the DHL in 0 or 4% dextran. Uncropped version of Western blot is shown in Appendix A. Abbreviations: DAPI, 4′,6-diamidino-2-phenylindole; HLA, human leukocyte antigen; GAPDH, Glyceraldehyde-3-phosphate dehydrogenase.

**Figure 2 ijms-22-10067-f002:**
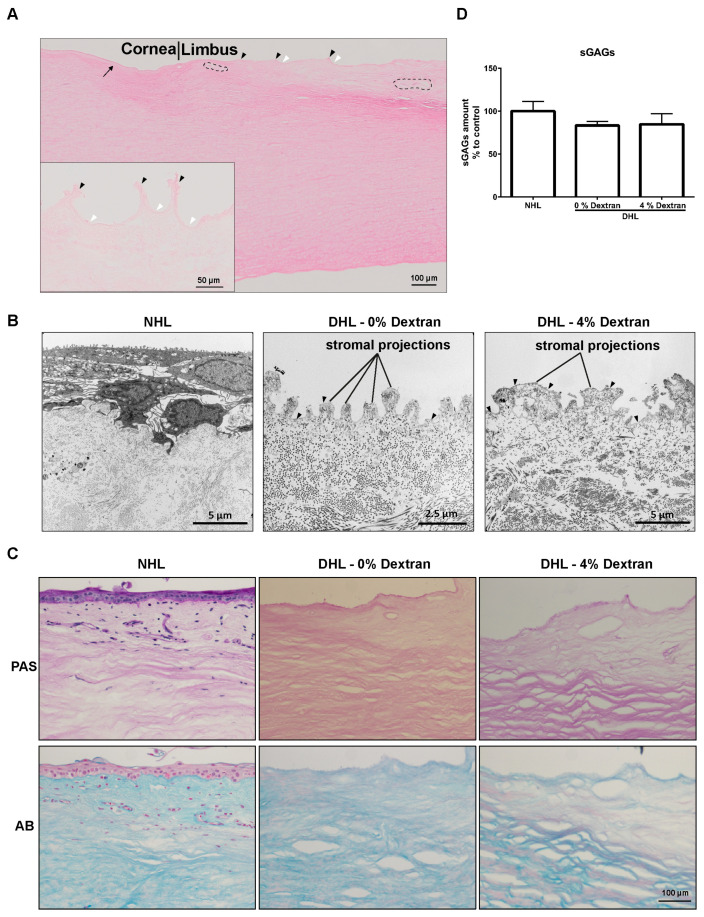
Structural and biochemical composition of the decellularized human limbus (DHL). (**A**) The hematoxylin and eosin staining of DHL scaffold showing the Bowman’s layer (arrow), connective tissue protrusions (black arrowheads), invaginations (white arrowheads), and vascular gaps (dashed circles). Insight showing the stromal projections (black arrowheads) and invaginations (white arrowheads). (**B**) The transmission electron microscopy (TEM) micrographs of the NHL and DHL in 0 or 4% dextran; the anterior surface of limbus showing cells on the NHL but no cells on the DHL; DHL scaffolds showing the stromal projections and the regular arrangement of collagen fibrils similar to NHL. (**C**) Histological evaluation of extracellular matrix content by periodic acid Schiff (PAS) and alcian blue (AB) on DHL (with 0 or 4% dextran) compared with NHL. (**D**) Sulfated glycosaminoglycan (sGAG) content of DHL in 0 or 4% dextran in comparison to NHL. The graph represents a percentage (%) of sGAG content in limbal scaffolds and data are expressed as means ± S.E.M. (*n* = 5).

**Figure 3 ijms-22-10067-f003:**
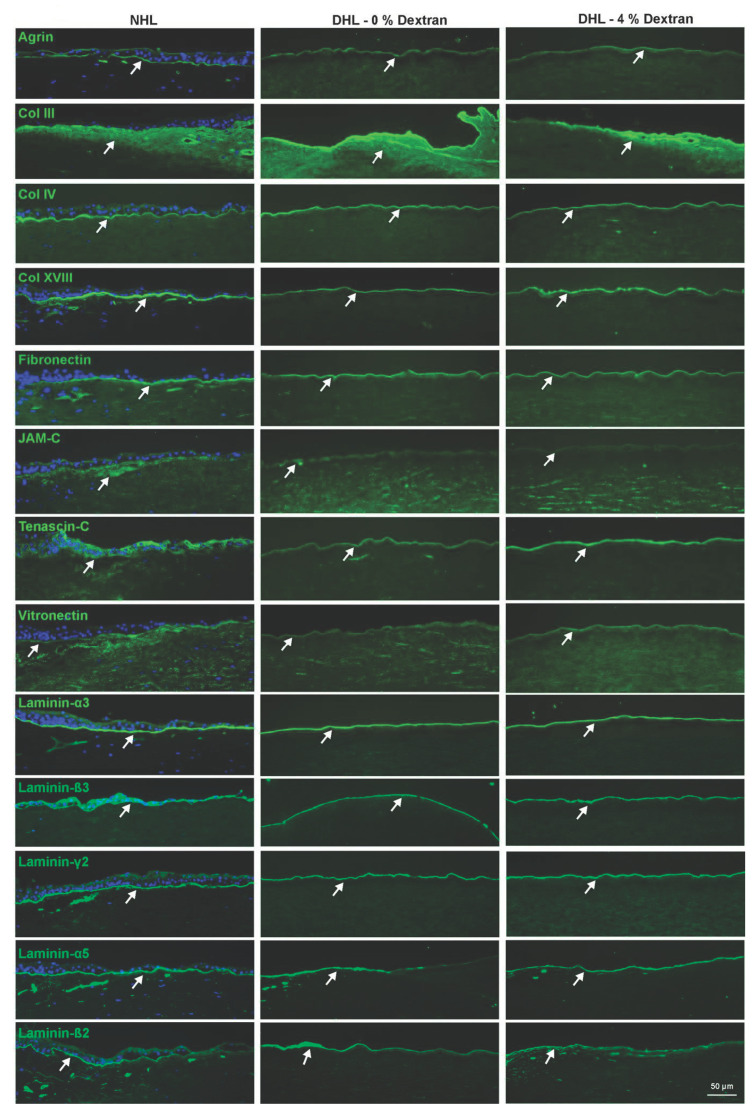
Extracellular matrix (ECM) composition of decellularized human limbus (DHL). Immunostaining analysis of various ECM and basement membrane (BM) molecules on the DHL (0 and 4% dextran) compared to the normal human limbus (NHL). Arrows indicate the expression of ECM and BM molecules. Nuclei counterstained with 4′,6-diamidino-2phenylindole (blue). Abbreviations: Col, Collagen; JAM-C, junctional adhesion molecule C.

**Figure 4 ijms-22-10067-f004:**
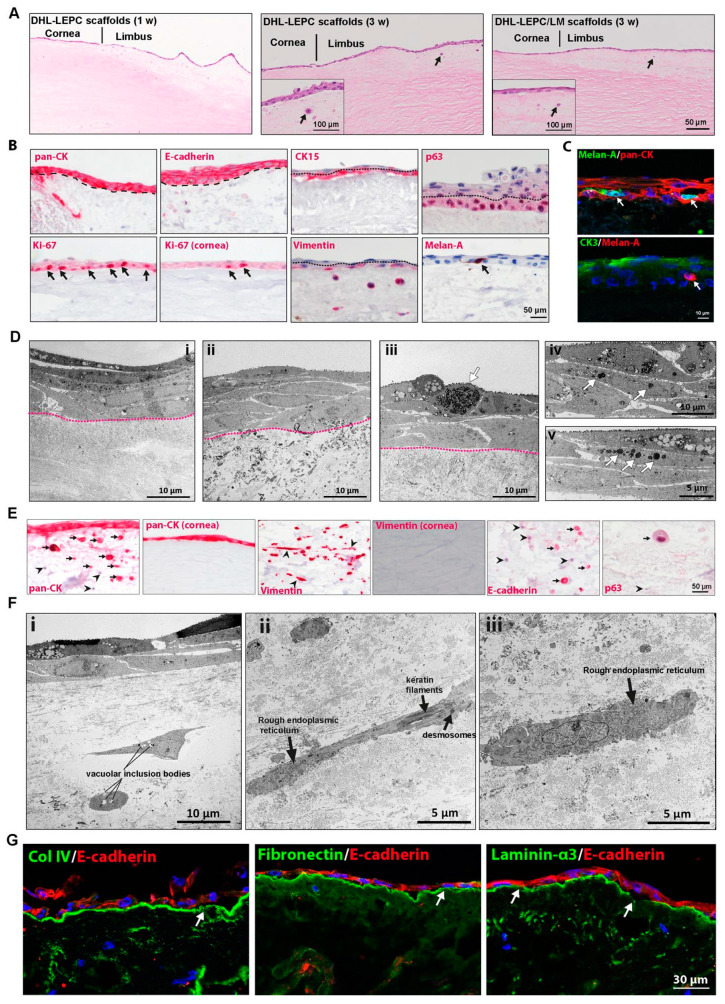
In vitro recellularization of the decellularized human limbus (DHL). (**A**) Light microscopic analysis of DHL–LEPC scaffolds showing cell monolayers after 1 week (1 w); after 3 weeks in culture, DHL–LEPC and DHL–LEPC/LM scaffolds showing stratified epithelium (3 w) and intrastromal invasion of epithelial-like cells (arrow) by hematoxylin and eosin staining (H&E) on the anterior surface. Insight showing the stratified epithelium and intrastromal invasion of epithelial-like cells (arrow). (**B**) Immunohistochemical staining on paraffin sections showing cytokeratin (pan-CK) expression and E (epithelial)-cadherin (magenta) in all human limbal epithelial cells (the dashed line separated the epithelium and stroma); CK15, p63, and vimentin (magenta) at basal layers of epithelial cells (dotted line separates basal and suprabasal cells); Ki-67^+^ cells at basal layers of epithelial cells in both limbal and corneal region (arrows); and Melan-A^+^ melanocytes (magenta, arrow) at basal layers of the epithelium. (**C**) Double immunofluorescence staining of DHL–LEPC/LM scaffolds showing the expression of cytokeratin (pan-CK, red) in all epithelial cells; CK3 expression (green) in suprabasal cells; and Melan-A^+^ melanocytes associated with basal cells (arrows). Nuclei stained with DAPI. (**D**) Transmission electron microscopy (TEM) micrographs of DHL–LEPC (**i**) and DHL–LEPC/LM (**ii**) scaffolds showing the multilayered epithelium (dotted line separates epithelium and stroma); darkly pigmented melanocyte in the epithelial layer of DHL–LEPC/LM scaffolds (**iii**); the presence of melanosomes in the cytoplasm of epithelial cells (**iv** and **v**, arrows). (**E**) Immunohistochemical staining of recellularized scaffolds showing PanCK^+^, E-cadherin^+^, and p63^+^ cells present in the limbal stroma, whereas no invasion of LEPC occurred in the corneal region. Pan-CK^−^, E-cadherin^−^ cells also present in the stroma; all cells in the stroma stained for vimentin and few with slender morphology (arrowheads). (**F**) TEM micrographs showing epithelial-like cells by their roundish shape, round nuclei, and vacuolar inclusion bodies in the limbal stroma (**i**); Fibroblast-like cells in the limbal stroma showed a mixed epithelial (keratin filaments, desmosome)–fibroblastic (spindle shape, prominent cisterns of rough endoplasmic reticulum) phenotype (**ii**); Epithelial-like cells in the limbal stroma presented signs of a fibroblastic transformation, as indicated by prominent rough endoplasmic reticulum (**iii**). (**G**) Immunofluorescence staining of recellularized scaffolds showing intact fibronectin, Col IV, and Laminin-α3 expression with occasionally distinct gaps (arrows). Nuclei stained with DAPI. Abbreviations: LEPCs, limbal epithelial progenitor cells; LMs, limbal melanocytes; pan-CK, pan-cytokeratin; CK15, cytokeratin 15; CK3, cytokeratin 3; DAPI, 4′,6-diamidino-2-phenylindole.

**Figure 5 ijms-22-10067-f005:**
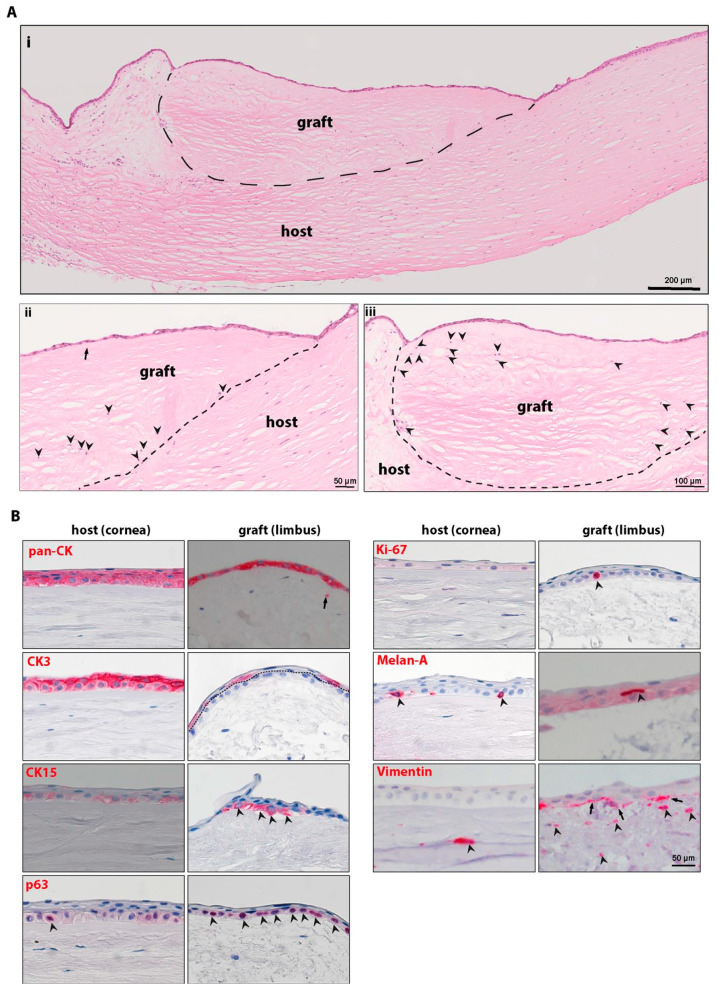
Ex vivo transplantation of decellularized human limbus (DHL). (**A**) Three weeks after transplantation, histological analyses by hematoxylin and eosin (H&E) staining show complete epithelialization of graft tissue with stratification (**i**); graft with Bowman’s layer (arrow, **ii**) and the presence of stromal cells in the graft tissue (arrowheads, **ii** and **iii**). Dashed line marks the boundary between the graft and host tissue. (**B**) Immunohistochemical staining of paraffin sections showing the cytokeratin (pan-CK) expression on epithelial cells; CK3 expression in all epithelial cells of the host, whereas only suprabasal cells in the graft (dotted line separates the basal and superficial cells); p63^+^ cells in the graft and host tissue (arrowheads), and Ki-67^+^ cells at basal layers of epithelial cells of the graft tissue (arrowheads); Melan-A expression (magenta, arrowheads) in melanocytes at basal layers of epithelium in both host and graft tissue, and vimentin expression in the basal cells of the epithelium (graft tissue, arrow) and stromal region (arrowheads) in both host and graft tissue. Abbreviations: pan-CK, pan-cytokeratin; CK3, cytokeratin 3; CK15, cytokeratin 15.

**Table 1 ijms-22-10067-t001:** Organ-cultured corneal sample details.

Corneal Samples	Number	Donor Age (years) *	Cultivation Duration (days) *
Total	136	69.1.2 ± 12.7 (25–96)	39.8 ± 11.2 (8–79)
- **DM^−^**	70	68.4 ± 11.2 (44–88)	38.0 ± 7.6 (22–59)
- **DM^+^**	37	79.8 ± 12.6 (51–96)	41.3 ± 9.7 (8–79)
- **After PK**	29	59.1 ± 14.5 (25–84)	40.3 ± 16.4 (26–73)

* All values are expressed as mean ± standard deviation (range). DM^−^—remnant tissue after Descemet’s membrane endothelial cell keratoplasty; DM^+^—corneal buttons not suitable for transplantation; after PK—remnant tissue after penetrating keratoplasty.

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
