# Peer review of "A Decellularized Human Limbal Scaffold for Limbal Stem Cell Niche Reconstruction"

_ijms, 2021, doi:10.3390/ijms221810067_

Round 1
Reviewer 1 Report
This manuscript reported the preparation and characterization of decellularized corneas and also reported the effects of this decellularized corneas on cells in vivo. Overall, the manuscript was well-written and the results seemed to support the conclusion.
However, I was wondering what the differences between this manuscript and their previous paper (Polisetti N et al., Sci Rep. 11, 2992, 2021, Ref 21). Very similar results were shown in their previous report and a few supporting data were added in this manuscript. The claims and conclusions of these two papers seemed to be almost similar. So, I felt that the manuscript did not have any progresses from their previous report. If the authors think it is new, they should described the differences more clearly in Introduction part.
For above reason, I cannot recommend this manuscript for the publication.
Author Response
Response to Reviewer 1
Point 1: However, I was wondering what the differences between this manuscript and their previous paper (Polisetti N et al., Sci Rep. 11, 2992, 2021, Ref 21). Very similar results were shown in their previous report and a few supporting data were added in this manuscript. The claims and conclusions of these two papers seemed to be almost similar. So, I felt that the manuscript did not have any progresses from their previous report. If the authors think it is new, they should described the differences more clearly in Introduction part.
Response 1: We thank the reviewer for raising this important point. Indeed, the pure decellularization protocols used in the previous publication and the current manuscript are similar and we failed to sufficiently clarify the different foci of the two papers.
The previous publication focused on decellularization of transparent central corneal tissue to produce scaffolds for repair of central corneal defects (Polisetti et al., 2021, ref. 24). In contrast, the current manuscript aims to provide a scaffold for limbal stem cell transplantation with the long-term goal to treat patients suffering from structural damage in the limbal stem cell niche. In the current work, we also decellularized the whole cornea in a first step to minimize possible handling-related tissue damage as reported by Isidan et al. (2021, ref. 37). However, in a second step, the limbo-corneal region was dissected by trephination and used in all subsequent analyses.
The limbal stem cell niche is very different from the central cornea studied in the previous report as it is endowed with crypt structures, fine stromal protrusions, a particular ECM composition and a basement membrane composition which regulates LEPC homeostasis (Goldberg MF et al., 1982, ref. 2; Polisetti et al., 2015, ref. 40; Shortt AJ et al., 2007, ref. 3; Schlotzer-Schrehardt et al., 2007, ref. 25). In light of these major structural and biological distinctions, we strongly believe that the effects of decellularization need to be tested with a distinct focus on the limbal tissue. The resulting data addressing a different region are therefore also novel and different from the previous report.
We have amended the manuscript to better clarify this issue.
Additional clarifications:
In addition to the above comments, minor changes have been made in the methodology and the results sections for clearer presentation.
We have added a supplementary figure 2 & 3 to allow for better understanding of the tissue preparation.
Various points have been added to the discussion to correlate the present findings with clinical significance.
References:
Polisetti, N.; Schmid, A.; Schlötzer-Schrehardt, U.; Maier, P.; Lang, S.J.; Steinberg, T.; Schlunck, G.; Reinhard, T. A Decellularized Human Corneal Scaffold for Anterior Corneal Surface Reconstruction. Sci. Rep. 2021, 11, 2992, doi:10.1038/s41598-021-82678-3.
Isidan, A.; Liu, S.; Chen, A.M.; Zhang, W.; Li, P.; Smith, L.J.; Hara, H.; Cooper, D.K.C.; Ekser, B. Comparison of Porcine Corneal Decellularization Methods and Importance of Preserving Corneal Limbus through Decellularization. PloS One 2021, 16, e0243682, doi:10.1371/journal.pone.0243682.
Goldberg, M.F.; Bron, A.J. Limbal Palisades of Vogt. Trans. Am. Ophthalmol. Soc. 1982, 80, 155–171.
Polisetti, N.; Zenkel, M.; Menzel-Severing, J.; Kruse, F.E.; Schlötzer-Schrehardt, U. Cell Adhesion Molecules and Stem Cell-Niche-Interactions in the Limbal Stem Cell Niche. Stem Cells Dayt. Ohio 2016, 34, 203–219, doi:10.1002/stem.2191.
Shortt, A.J.; Secker, G.A.; Munro, P.M.; Khaw, P.T.; Tuft, S.J.; Daniels, J.T. Characterization of the Limbal Epithelial Stem Cell Niche: Novel Imaging Techniques Permit in Vivo Observation and Targeted Biopsy of Limbal Epithelial Stem Cells. Stem Cells Dayt. Ohio 2007, 25, 1402–1409, doi:10.1634/stemcells.2006-0580.
Schlötzer-Schrehardt, U.; Dietrich, T.; Saito, K.; Sorokin, L.; Sasaki, T.; Paulsson, M.; Kruse, F.E. Characterization of Extracellular Matrix Components in the Limbal Epithelial Stem Cell Compartment. Exp. Eye Res. 2007, 85, 845–860, doi:10.1016/j.exer.2007.08.020.

Reviewer 2 Report
An elegant set of experiments that confirm the viability of using decellularised limbal tissue to improve limbal stem cell transplant.
The discussion would be enhanced with a more comprehensive discussion of the factors affecting graft survival, including inflammation and rejection as well as loss of the niche. Some analysis of the suspected role of the niche in graft survival and any clinical evidence of its protective effect would make the case for a clinical trial more compelling
Author Response
Response to Reviewer 2
Point 1: An elegant set of experiments that confirm the viability of using decellularised limbal tissue to improve limbal stem cell transplant. The discussion would be enhanced with a more comprehensive discussion of the factors affecting graft survival, including inflammation and rejection as well as loss of the niche. Some analysis of the suspected role of the niche in graft survival and any clinical evidence of its protective effect would make the case for a clinical trial more compelling
Response 1: We thank the reviewer for the kind appreciation of our work. We also thank the reviewer for the suggestions to improve the quality of the manuscript. The following points have been added to the discussion:
“In cases of allogenic limbal transplantation, immunoresponse-related damage to the transplanted limbal tissue is a crucial factor for chronic graft failure and most patients require immunosuppression[15]. Therefore, there is an unmet need for developing a scaffold that more closely mimics the limbal niche and exhibits very low immunogenicity to better maintain LEPC during in-vitro cell expansion and minimize immune response in the host”.
“The lack of nuclear and cellular material in decellularized tissues reduces the antigenicity of the limbal scaffolds and could therefore provide nonimmunogenic scaffolds for LEPC transplantation to treat LSCD”.
“The limbal tissue has a specialized BM and ECM protein composition , which contributes to maintenance of LEPC phenotype and influences the amount and composition of cytokines and growth factors acting on LEPC[25,39,40].”
“It has been reported that ex-vivo expanded allogenic LEPC were not observed on the ocular surface 3 months after transplantation[45]. In contrast, allogenic cells were evident for up to 3 years after a keratolimbal allograft[46]. These findings illustrate the importance of LEPC niche components for long-term LEPC survival. The preservation of limbus-specific ECM (Col III, TNC, JAMC) and BM components (LN-α5 and -β2) observed in the current study is suited to support adhesion, proliferation, and long-time survival of LEPC.”
“It has been stated that a certain number of p63+ (more than 3%) in the donor tissue required for successful regeneration of the host corneal surface[10]. Our data showed the preservation of p63+ LEPC throughout the limbal basal layer (Fig.4B & Fig. 5B), suggesting the potential application of these scaffolds for corneal surface regeneration.”
“The appearance of LM on transplanted limbal scaffolds supports the recent clinical finding of reappearance of limbal pigmentation after 8 months of a simple limbal epithelial transplant[50]. So, LMs acts as emerging key players in niche regulation of LEPC especially in wound healing[31].”

Reviewer 3 Report
The manuscript reports the formation of decellularized human limbal scaffold for cornea/limbal tissue regeneration. The main issue of this manuscript is the high degree of similarity to the previous paper by the same authors (ref 21). The experimental findings are very much the same, esp. the recellularization and the ex vivo transplantation parts. While the presentation is good, I find it difficult to accept this manuscript as there is little novelty and scientific value-add in this report.
I would suggest to re-position this manuscript and come in using a new angle. Perhaps a side-by-side comparison with the current standard-of-care fibrin gel could be made to provide some fresh insights and interesting findings.
Author Response
Please see the attachement

Round 2
Reviewer 1 Report
The authors answered and I understood your purpose and the differences from previous study. So, I recommend to publish this manuscript as it is.
Author Response
We thank the reviewer for the kind acceptance of our purpose of the study. We also thank the reviewer for the suggestions, which improved the quality of the manuscript a lot.
Reviewer 3 Report
The added data and explanation has provided a better understanding of the differences between the 2 studies.
I would be ok to accept the manuscript after all the revisions in the text are highlighted and in particular to include the response to point 1 in the introduction section to clearly clarify the differences.
